# Maternal socioeconomic status and infant feeding practices underlying pathways to child stunting in Cambodia: structural path analysis using cross-sectional population data

Chloe Mercedes Harvey  ,[1] Marie-Louise Newell,[2,3] Sabu Padmadas  [4]

¹Independent Consultant, Bangkok, Thailand
²Faculty of Medicine, School of Human Development and Health, Global Health Research Institute, University of Southampton, Southampton, UK
³Faculty of Health Science, School of Public Health, University of the Witwatersrand, Johannesburg-Braamfontein, Gauteng, South Africa
⁴Social Statistics and Demography, University of Southampton, Southampton, UK

**Correspondence to**
Dr Chloe Mercedes Harvey;
chloe.mercedes.harvey@gmail.com

## ABSTRACT

**Objectives** To identify and investigate complex pathways to stunting among children aged 6–24 months to determine the mediating effects of dietary diversity and continued breast feeding on the association between socioeconomic factors and child stunting.

**Design, setting and participants** We analysed the most recent cross-sectional Demographic and Health Survey data from Cambodia (2014). We applied structural path analysis on a sample of 1365 children to model the complex and inter-related pathways of factors determining children's height for age. Explanatory variables included a composite indicator of maternal employment, household wealth, maternal education, current breastfeeding status and dietary diversity score. Results are presented both in terms of non-standardised and standardised coefficients.

**Outcome measure** The primary outcome measure was height-for-age Z-scores as a continuous measure.

**Results** Findings suggest that children's dietary diversity and continued breast feeding mediate the association between socioeconomic status and children's height. While there was no significant direct effect of maternal education on children's height, results suggested significant indirect pathways through which maternal education effects children's height; operating through household wealth, maternal employment, dietary diversity and continued breastfeeding status (p<0.001). Most notably, 41% of the effect of maternal employment on children's height was mediated by either dietary diversity or continued breast feeding.

**Conclusion** We provide evidence to support targeted nutrition interventions which account for the different ways in which underlying socioeconomic factors influence infant and young child feeding practices, and the potential impact on child nutritional status.

## STRENGTHS AND LIMITATIONS OF THIS STUDY

⇒ Application of a causal hypothetical model to explain possible pathways to children's linear growth.
⇒ Hypothetical models that can be applied to similar geographical contexts.
⇒ The cross-sectional nature of Demographic and Health Survey data requires that results be interpreted with some caution.
⇒ The theoretical model was built iteratively based on prior theory and thus is not an exhaustive representation of the pathways to children's height.

## INTRODUCTION

Across Southeast Asia, stunting in children under 5 years of age continues to pose a critical public health concern, with most countries in the region experiencing prevalence levels above 30%.[1] In Cambodia, more than a third of children under the age of 5 years are stunted[1]; stunting has been shown to be associated with distinct socioeconomic inequality, with suboptimum feeding practices especially after 6 months of age, micronutrient deficiencies and recurrent infections.[2 3]

Impaired linear growth has adverse long-term health consequences for cognitive and immune system development, and is associated with increased risk of overweight and comorbidities in later life.[2 4 5] Evidence shows that childhood stunting also has intergenerational implications, as stunting in pregnant women may lead to intrauterine growth restriction in her offspring,[6] thus continuing the vicious cycle of undernutrition. Interventions to tackle undernutrition in Southeast Asia have not been a top priority for governments and other stakeholders,[7] suggesting the need for strengthening further scientific evidence to understand the causal pathways of undernutrition, to identify appropriate strategies for policy and programme interventions.

Biobehavioural and socioeconomic factors associated with children's height for age operate at varying levels of causation.[2] Limited dietary diversity has previously been reported to be significantly associated with impaired linear growth,[8] whereas the role

of continued breast feeding beyond 6 months remains inconclusive, with some studies even suggesting a negative association.[9] [10] However, it is not clear whether early feeding practices are directly and independently associated with height for age or whether they mediate the association between distal (underlying) factors and stunting.

Studies in Cambodia and other low and middle-income countries have consistently shown maternal education, household wealth and maternal employment to be associated with inequalities in stunting prevalence.[2] [11–15] These three socioeconomic factors have also been associated with stark differentials in breastfeeding practices and dietary diversity within the Southeast Asian context.[16] However, associations underlying improved maternal socioeconomic status, continued breast feeding and dietary diversity are paradoxical, as improvements in socioeconomic status (such as income, employment and education) are congruent with declines in breast feeding, whereas improved socioeconomic status with increasing household income is positively associated with improved dietary diversity. Previous research has also indicated significant differentials in child nutritional status according to urban/rural residence[2] [15]; however, the intraurban and intrarural differentials are rarely considered, despite research suggesting starker socioeconomic inequalities within urban areas in low and middle-income country settings.[17]

Most cross-sectional studies to date focusing on factors associated with linear growth have examined children's height for age as a dichotomous outcome, expressed as stunting. Evidence on the complex and inter-related nature of factors associated with a continuous measure of children's height for age is limited, including modelling techniques that account for dynamic interactions between possible explanatory variables, to disentangle direct effects from indirect effects with mediating variables.[18–20] To date, no such study has focused on the Southeast Asian region, including Cambodia, where evidence-based interventions are critical for tackling the population burden of child undernutrition.

Using cross-sectional data from the most recent Cambodian Demographic and Health Survey (CDHS) in Cambodia, we identify and investigate theoretical pathways to stunting among children aged 6–24 months, when breastfeeding and dietary practices are most important for growth. We use structural path analysis to examine two hypotheses: (1) socioeconomic factors are directly associated with dietary diversity and continued breast feeding; and (2) dietary diversity and continued breast feeding beyond 6 months of age mediate the association between socioeconomic factors and children's height for age, partially explaining the association between underlying socioeconomic factors and the outcome (children's height).

## METHODS

### Data

We used the most recent Demographic and Health Survey (DHS) data from Cambodia,[20] which adopted a two-stage stratified sampling by separating rural and urban areas. The survey interviewed 17 578 women aged 15–49 years from 15 825 households, which included a sample of women from 5667 urban and 11 911 rural areas. Anthropometric measurements, child health and healthcare data were collected for only those births that occurred 5 years preceding the survey. All eligible mothers were asked about the breastfeeding status of their children under 36 months at the time of survey visit; 24-hour recall of additional liquids and solid foods was collected for the youngest child <36 months.

The sample of interest for this study was the youngest, living singleton child, between the ages of 6 and <24 months at the time of survey, living in the same household as their mother. As per CDHS (2014) sampling design, anthropometric measurements were only taken in two-thirds of households, and after accounting for missing or implausible anthropometric information, a total of 1365 Cambodian children between the ages of 6 and <24 months were selected (online supplemental table S1). The children's file contains the records for every child born in the 5 years preceding the survey for women aged 15–49 years, who were selected randomly from the sampled households within each cluster.

The response rate for the women's survey from which information about the children is collected was 98%. Additional information on the CDHS sampling design and survey methodology is reported elsewhere (CDHS, 2014). Item non-response for the selected explanatory variables was minimal, and the decision was made to exclude cases with missing data at random from the final analysis (<0.5% of total cases).

No additional data are available.

### Dependent variable

Anthropometric data were collected by trained interviewers using internationally standard instruments. Height measurements were taken using a SECA measuring board.[20] For children aged less than 24 months, recumbent length measurements were taken, and for older children standing height was measured.[21]

Children's height-for-age Z-scores (HAZ) were calculated by the DHS using the WHO Child Growth Standards.[22] Stunting was defined as HAZ below −2 SD from the mean of the reference population and severe stunting as below −3 SD.[22] We used HAZ as a continuous measure, assuming that explanatory variables have a varying effect on the respective Z-score distribution.

### Explanatory variables

Selection of explanatory variables was based on previous research on the association between infant and young child feeding practices and child stunting, as well as two conceptual frameworks which depict the pathways

through which different factors may affect child height/growth at different levels.[23 24] Children's dietary diversity and breast feeding at time of survey were selected as the two main proximate factors in the hypothetical structural path models. This was on the basis of evidence from the Cambodian DHS that suggests the main reason for Cambodian children not meeting the minimum acceptable diet was due to not meeting the recommended minimum dietary diversity (MDD; consumption of four or more food groups in the previous 24 hours) but not due to meal frequency.[20] Inclusion of continued breast-feeding status separately is important as while it is not included as a food group in the original WHO definition of MDD-7, it is recommended as an additional source of nutrients during the complementary feeding period.[25]

Terciles of dietary diversity were computed by age in months, based on the number of food groups consumed by children in the 24 hours preceding the survey,[26] to account for the gradual introduction of complementary foods.[27] Thus, children were regarded as having a low, middle or high diversity, relative to children of the same age. The definition of the terciles varies by age; online supplemental table S2 presents the distributions of the dietary diversity terciles by age.

Breast feeding was categorised as a dichotomous variable as whether the child was being currently breast fed or not at child's age or time of survey.

The distal (underlying) factors included maternal education, household wealth and maternal employment. Maternal education reflected the highest level of completed education and was categorised into no education, primary, secondary and higher. Quintiles of household wealth were computed separately for urban and rural areas by DHS using principal component analysis, to ensure the representation of the poorest of the poor households, especially in rural areas.[28]

As children's nutritional status and feeding practices may be determined by a combination of different aspects and types of maternal employment,[29 30] we created a composite indicator measuring maternal participation in the labour force based on an indicator computer by Sebayang *et al*.[31] Using principal component analysis, we incorporated five aspects of maternal employment: (1) working status over the past 12 months; (2) type of employment, whether self-employed or employed by family; (3) type of occupation; (4) type of earnings; and (5) seasonality of work. Three distinctive categories of maternal employment were identified: not working, low level of participation, high level of participation, which was further dichotomised into a binary variable: (0) not working/low participation; (1) high participation.

## Statistical analysis

Bivariate analyses included adjusted Wald tests for joint hypothesis testing to assess preliminary associations between explanatory variables and HAZ (online supplemental table S3). Statistical significance was considered at p<0.05 level.

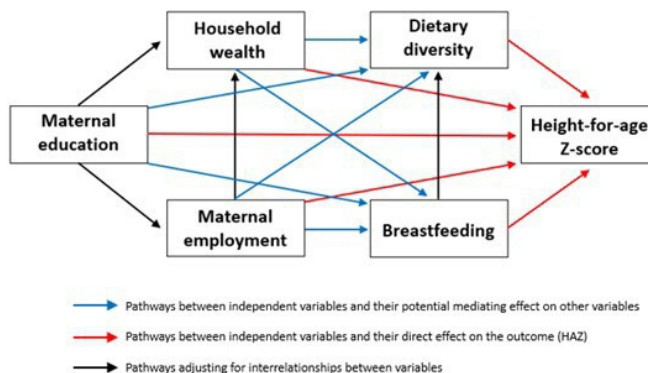

**Figure 1** Conceptual model of pathways between selected socioeconomic factors, dietary diversity, breast feeding and height-for-age Z-scores (HAZ).

Analysis was conducted using Stata/SE V.15.1.[32] All descriptive and exploratory analyses accounted for the complex survey design of the CDHS using the appropriate sample weights.[20]

Structural path analysis is the structural component of structural equation modelling and involves the modelling of observed variables only.[33] Path analysis facilitates the study of direct and indirect effects of multiple independent and dependent variables simultaneously, based on a generalised multiequation framework for examining multiple hypotheses about interdependencies between these variables.[34 35] The first step of analysis involved creating a hypothetical 'causal' pathway diagram between the independent variables of interest and the outcome HAZ (figure 1). The hypothesised model was built iteratively based on prior theory and research, and therefore not an exhaustive representation of the pathways to children's height for age. It was selected as the appropriate methodology for this analysis as it allows for the identification of indirect and direct mechanisms through which socioeconomic factors influence height for age, where dietary diversity and breast feeding may act as mediators in these associations.

Structural equation modelling assumes linearity of observed variables, and the assumption of linear normality among pseudocontinuous variables was tested with likelihood ratio tests and Bayesian information criterion/Akaike information criterion diagnostics.[33] Most independent variables showed evidence of linearity, justifying their inclusion in the model as continuous variables. However, this was not the case for maternal employment as a continuous independent variable, and we thus used the dummy variable to adjust for high level of participation in employment, as this was the category of interest for this study.

## Assessment of model fit

Structural path analysis was undertaken using the SEM programme in Stata/SE V.15.1. All models were estimated using maximum likelihood with missing values. Goodness of fit of the data to the models was assessed using the root mean square error of approximation (RMSEA <0.06) and

the comparative fit index (CFI ≥0.95 for acceptance).[36] Both non-standardised and standardised coefficients are presented, but only standardised results are interpreted and presented on path diagrams (non-standardised coefficients use the raw/real-life units of each variable and standardised coefficients convert the coefficients into comparable units). Standardised coefficients are interpreted in terms of SD differences.[34] To address the second hypothesis of this study that suggests dietary diversity and breast feeding are mediators in the association between socioeconomic status and children's HAZ, standardised indirect effects were calculated.[33]

## Patient and public involvement

Secondary data were used for this analysis. Therefore, no patient consent was needed.

## RESULTS
### Sample characteristics

Approximately one-third of children were aged between 6 and 12 months and two-thirds were aged 12–23 months (table 1), with 52% boys and 48% girls; 6% of children were born with a low birth weight (<2.5 kg). Just over two-fifths (44%) of children were reported to have had morbidity, either a fever, respiratory condition or diarrhoea, in the 2 weeks prior to interview. Five per cent of mothers were of short stature (<145 cm); 16% had low (<18.5) and 15% had high (≥25.0) body mass index. Most mothers had completed primary-level education (56%), and only 12% reported no formal education. Approximately a third of women reported that they had not worked in the year prior to survey, a third of women had low level and the remainder high level of participation in employment. One in four of the children were from the poorest households and most households were located in rural areas (86%). About 70% of children aged 6–23 months were still being breast fed at the time of survey; 34% had been fed from a bottle. Approximately 43% of children had a low dietary diversity, but 73% of children consumed the recommended meal frequency.

### Results of structural path analysis
#### Underlying socioeconomic factors

Figure 2 represents the full structural path model; the standardised and unstandardised coefficients are presented in online supplemental table S4. Overall, the fit of the model according to RMSEA (<0.05) and CFI (>0.950) was good, and the model explained approximately 44% of the variance in HAZ ($R^2$=0.44).

Maternal education was significantly and positively associated with dietary diversity. One SD increase in the level of maternal education was associated with a 0.07 SD increase in dietary diversity (β=0.07; p=0.023), but with no significant effect on breastfeeding status. Similarly, a 1 SD increase in household wealth was associated with a 0.18 SD increase in dietary diversity tercile (β=0.18; p<0.001), but a 1 SD increase in household wealth was associated

with –0.09 SD decrease in breast feeding. A high level of maternal employment was associated with a positive but insignificant effect on dietary diversity tercile, but statistically significantly negatively associated with breastfeeding status (β=–0.07; p=0.008). Further analysis revealed that this effect on breastfeeding status was confirmed among younger age groups, but not in the age group of 18–23 months (online supplemental table S5).

### Dietary diversity and continued breast feeding beyond 6 months of age as mediating factors

Both hypothesised mediating variables—dietary diversity tercile and breastfeeding status—were independently and significantly associated with child HAZ, although associations were in the opposite direction. One SD increase in dietary diversity was associated with a 0.06 SD increase in HAZ (p=0.017), whereas continued breast feeding was associated with a –0.13 decrease in HAZ (p<0.001) (figure 2 and online supplemental table S4). The only socioeconomic factor with a significant independent association with HAZ was household wealth, where a 1 SD increase in household wealth was associated with a 0.06 SD increase in HAZ (p=0.045).

To assess the role of dietary diversity and breast feeding as mediators in the association between socioeconomic factors (maternal education, household wealth, maternal employment (high)) and HAZ, indirect pathways were examined (table 2). In terms of maternal education, the standardised effects presented a case of 'inconsistent mediation',[37] where the direct and indirect effects do not have the same consistent sign. Furthermore, while the indirect effect is significant (p<0.001), the direct effect is not, and hence the indirect effect is larger than the total effect, meaning that the effect of maternal education on HAZ is fully attributed to the indirect path which operates through household wealth, maternal employment, dietary diversity and breastfeeding status.

Although there was no direct effect of maternal employment on HAZ, statistically significant indirect effects suggested that 41% (0.039/0.094=0.41) of the effect of maternal employment on HAZ was mediated by either dietary diversity or breastfeeding status.

Household wealth was the only underlying (socioeconomic) factor with a statistically significant direct effect on HAZ, with 71% of the total effect on HAZ estimated to be direct (0.057/0.080=0.71), after controlling for dietary diversity and breastfeeding status, with the indirect effect of household wealth on HAZ an estimated 28% (0.022/0.080=0.28).

Further exploration of the specific indirect pathways between underlying (socioeconomic) factors and HAZ, through mediating variables of dietary diversity and breast feeding, showed evidence of three indirect pathways that were statistically significant (online supplemental table S6). High level of maternal employment had a small but significant standardised effect on HAZ, mediated by breastfeeding status of 0.01, z=2.30, p=0.022. Household wealth also had a small but statistically significant

**Table 1** Summary characteristics of children 6–23 months from the Cambodian DHS (2014) (n=1381*)

| Indicator | | % | n | Mean HAZ | SD | 95% CI for mean | |
|---|---|---|---|---|---|---|---|
| | | | | | | Lower bound | Upper bound |
| **Child characteristics** | | | | | | | |
| Age (months) | 6–8 | 17.5 | 242 | −0.55 | 1.25 | −0.75 | −0.35 |
| | 9–11 | 15.1 | 208 | −0.92 | 1.48 | −1.14 | −0.70 |
| | 12–17 | 33.5 | 463 | −1.41 | 1.23 | −1.54 | −1.28 |
| | 18–23 | 33.9 | 468 | −1.46 | 1.32 | −1.60 | −1.31 |
| Sex | Male | 52.0 | 718 | −1.28 | 1.34 | −1.39 | −1.16 |
| | Female | 48.0 | 663 | −1.12 | 1.35 | −1.24 | −1.0 |
| Birth weight (kg) | <2.5 | 6.0 | 83 | −1.48 | 1.50 | −1.85 | −1.11 |
| | 2.5<4.0 | 83.4 | 1152 | −1.16 | 1.33 | −1.25 | −1.07 |
| | ≥4.0 | 3.4 | 46 | −0.82 | 1.10 | −1.17 | −0.46 |
| | Missing/not weighed | 7.2 | 100 | −1.63 | 1.38 | −1.91 | −1.36 |
| Morbidity | No | 56.5 | 781 | −1.16 | 1.40 | −1.27 | −1.05 |
| | Yes | 43.5 | 601 | −1.26 | 1.28 | −1.39 | −1.13 |
| **Maternal characteristics** | | | | | | | |
| Height (cm) | Normal stature (≥145) | 94.8 | 1308 | −1.16 | 1.34 | −1.25 | −1.08 |
| | Short stature (<145) | 5.2 | 71 | −1.90 | 1.36 | −2.21 | −1.59 |
| BMI | <18.5 | 15.6 | 215 | −1.50 | 1.22 | −1.68 | −1.32 |
| | 18.5–24.99 | 69.2 | 952 | −1.17 | 1.39 | −1.27 | −1.06 |
| | ≥25.0 | 15.2 | 209 | −1.09 | 1.24 | −1.31 | −0.86 |
| Highest education level | No education | 12.0 | 166 | −1.26 | 1.45 | −1.50 | −1.02 |
| | Primary | 55.6 | 768 | −1.28 | 1.27 | −1.40 | −1.16 |
| | Secondary/higher | 32.4 | 447 | −1.04 | 1.42 | −1.18 | −0.91 |
| Maternal employment | Not working | 30.5 | 421 | −1.19 | 1.22 | −1.34 | −1.04 |
| | Low | 37.5 | 517 | −1.40 | 1.42 | −1.54 | −1.26 |
| | High | 32.0 | 441 | −0.98 | 1.34 | −1.12 | −0.84 |
| **Household characteristics** | | | | | | | |
| Household wealth index | Poorest | 25.4 | 345 | −1.48 | 1.49 | −1.65 | −1.32 |
| | Poorer | 18.2 | 248 | −1.32 | 1.41 | −1.50 | −1.14 |
| | Middle | 19.9 | 271 | −1.23 | 1.18 | −1.41 | −1.05 |
| | Richer | 17.7 | 241 | −1.0 | 1.21 | −1.20 | −0.80 |
| | Richest | 18.8 | 256 | −0.86 | 1.17 | −1.04 | −0.67 |
| Urban/rural residence | Rural | 86.4 | 1193 | −1.25 | 1.23 | −1.34 | −1.16 |
| | Urban | 13.6 | 188 | −0.88 | 1.97 | −1.06 | −0.70 |
| Sex of household head | Male | 77.0 | 1064 | −1.20 | 1.33 | −1.29 | −1.10 |
| | Female | 23.0 | 317 | −1.21 | 1.39 | −1.40 | −1.02 |
| **IYCF characteristics** | | | | | | | |
| Breast feeding | No | 29.9 | 413 | −1.15 | 1.39 | −1.30 | −0.99 |
| | Yes | 70.1 | 968 | −1.22 | 1.33 | −1.32 | −1.12 |
| Bottle feeding | No | 66.0 | 911 | −1.27 | 1.33 | −1.38 | −1.16 |
| | Yes | 34.0 | 469 | −1.07 | 1.37 | −1.20 | −0.93 |
| Dietary diversity tercile | Low | 43.2 | 596 | −1.30 | 1.20 | −1.42 | −1.17 |
| | Middle | 33.8 | 467 | −1.29 | 1.46 | −1.44 | −1.13 |
| | High | 23.1 | 318 | −0.90 | 1.41 | −1.07 | −0.73 |

**Table 1** Continued

| Indicator | | % | n | Mean HAZ | SD | 95% CI for mean | |
| --- | --- | --- | --- | --- | --- | --- | --- |
| | | | | | | Lower bound | Upper bound |
| Minimum meal frequency | No | 27.4 | 378 | −1.32 | 1.38 | −1.48 | −1.16 |
| | Yes | 72.6 | 1003 | −1.16 | 1.33 | −1.25 | −1.06 |
| Minimum acceptable diet | No | 68.4 | 945 | −1.23 | 1.32 | −1.33 | −1.13 |
| | Yes | 31.6 | 436 | −1.13 | 1.41 | −1.29 | −0.98 |

*Sample size differs slightly from the sample used in the final models (n=1365) due to missing data for some children on explanatory variables.
BMI, body mass index; DHS, Demographic and Health Survey; HAZ, height-for-age Z-scores; IYCF, infant and young child feeding.

standardised effect on HAZ, mediated by both dietary diversity and breastfeeding status (0.01, $z$=2.13, p=0.033; 0.01, $z$=2.58, p=0.012 respectively).

## DISCUSSION
The current study extends previous research examining factors associated with child stunting, to present a robust statistical model, which reflects the complex interactions between variables associated with children's height for age. Structural path analysis allows for the replication of a theoretical framework with underlying and proximate factors, to examine both the indirect and direct effects of factors known to be associated with child stunting. The structural path model presented could be applicable to other low and middle-income country contexts and could examine whether our proposed pathways to stunting are consistent across geographical contexts.

Overall, the results of this study confirmed that improvements in socioeconomic status, such as maternal education, household wealth and participation in employment, were associated with an increase in children's dietary diversity, but a decrease in continued breast feeding beyond 6 months of age. The findings also suggested that dietary diversity and continued breast feeding mediate the associations between socioeconomic status and HAZ. High level of maternal employment had a small but significant association with HAZ that was mediated by breastfeeding status, while the significant association between

household wealth and HAZ was mediated by both dietary diversity and breastfeeding status. This suggests that interventions to improve child nutritional status should make due consideration for the underlying socioeconomic drivers which may influence infant and young child feeding practices. Women's employment condition is one such factor that has been found to influence infant feeding practices globally, especially early weaning among mothers returning to work.[38] Although the female labour force participation rate is relatively high in Cambodia, with 74% of women participating in the labour force,[39] Cambodia has one of the highest proportions of informally employed (93.1% among men and women),[40] which is often associated with a lack of access to maternity protection.[41]

The independent effect of maternal education on dietary diversity was small but significant, consistent with the findings of other studies conducted in low and middle-income countries,[42–45] in addition to studies which confirmed a strong association between increasing household wealth and improvements in dietary diversity.[15 46] The significant pathway between breastfeeding and dietary diversity demonstrated an additional, interesting observation, that continued breastfeeding beyond 6 months of age, was associated with a less diverse diet, even after adjusting for age. This could be indicative of the situation in poorer households where prolonged breast feeding may be practised to make up for the lack of complementary foods, as suggested from a study of feeding practices in Thailand.[9]

In light of the nutrition transition, which has been occurring in many low and middle-income countries, Popkin and Bisgrove[47] reported that duration of breast feeding was declining in urban areas as a result of the increased maternal education and employment opportunities that arise from urbanisation. Furthermore, the negative association between increasing household wealth and continued breast feeding is perhaps underscored by the increasing trend in use of breast milk substitutes among children aged over 6 months in Cambodia.[48] Although the Cambodian government adopted many aspects of the *International Code of Marketing of Breast-milk Substitutes*[49] in the Cambodian *Sub-Decree on Marketing of Products for Infant and Young Child Feeding (Sub-Decree 133),*[50] a recent pilot implementation of a monitoring and enforcement

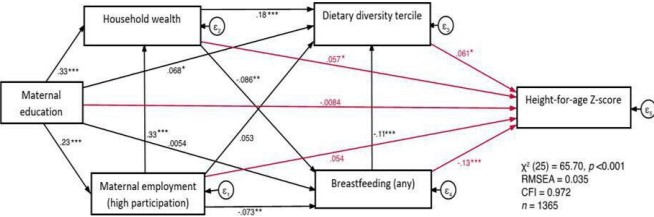

**Figure 2** Full structural path model measuring how socioeconomic factors are associated with children's height-for-age Z-scores (HAZ) directly and through mediators of dietary diversity and continued breast feeding. Standardised coefficients. Cambodia (2014). Model is adjusted for age, sex, birth weight, maternal stature and urban/rural residence. Root mean square error of approximation (RMSEA <0.05) and comparative fit index (CFI >0.95) indicate an overall good fit of the model. *P<0.05; **p<0.01; ***p<0.001.

**Table 2** Standardised coefficients of maternal education, household wealth, maternal employment and child height-for-age Z-scores, with correlated residuals for dietary diversity and breastfeeding status

| Outcome | Direct effect (β) | Indirect effect (β) | Total effect |
|---|---|---|---|
| HAZ | | | |
| Dietary diversity ->HAZ | 0.061* | na | 0.061* |
| Breast feeding ->HAZ | −0.129*** | −0.007 | −0.136*** |
| Maternal education ->HAZ | −0.008 | 0.051*** | 0.043 |
| Maternal employment (high) ->HAZ | 0.054 | 0.039*** | 0.094** |
| Household wealth ->HAZ | 0.057* | 0.022** | 0.080** |

The significance levels are for the unstandardised solution.
*P<0.05; **p<0.01; ***p<0.001.
HAZ, height-for-age Z-scores; na, not applicable.

system found significant non-compliance with this Sub-Decree at point of sales and in health facilities.[51]

Our study confirmed that dietary diversity and continued breast feeding mediate associations between socioeconomic status and child stunting; however, these findings were only significant for pathways originating from maternal employment and household wealth. The results of this study suggested a positive and direct association between Cambodian children's dietary diversity and HAZ, which has also been supported by other studies conducted in Cambodia.[52 53] Moreover, dietary diversity was a significant mediator of the association between household wealth and HAZ, suggesting that the positive effect of increasing household wealth can partially be explained by improved dietary diversity. This highlights the need to prioritise targeted interventions within the poorest households to improve dietary diversity to prevent child stunting. A further 71% of the total effect of household wealth on HAZ was found to be direct, suggesting the importance of other mediating factors not included in this model but known to be associated with child stunting, such as access to healthcare; water, sanitation and hygiene facilities; and immunisation status.

On the other hand, breast feeding after age 6 months was associated with a negative, direct effect on children's HAZ in Cambodia in the path model that adjusted for age of the child. This was also noted by Arimond and Ruel[27] in a study using Cambodian DHS data from 2000, and in a review of 13 studies by Grummer-strawn.[54] A negative effect of breast feeding beyond age 6 months on children's height-for-age status has been discussed by many researchers. One proposed explanation is reverse causality, whereby the decision to wean is related to mother's perceptions of size and stature of the child, hence malnourished children may be more likely to be breast fed for longer durations.[55] However, as Martin[56] points out, due to the cross-sectional nature of structural path analysis, it is not possible to account for the direction in the relationship between prolonged breast feeding and HAZ. In terms of the indirect effects, structural path analysis revealed that breast feeding was a small but significant mediator in the association between maternal employment and HAZ, as well as household wealth and HAZ.

## Strengths and limitations

This study used population-based data from the most recent nationally representative household survey from Cambodia, and our findings are generalisable to Cambodia. DHS data use standard instruments to measure height by professionally trained investigators, which ensures accuracy of anthropometric measurements; however, the way in which continued breast feeding and dietary diversity are reported using current status data (from a 24-hour recall questionnaire) may introduce measurement bias, although this would apply equally to stunted and non-stunted children. Although the 2014 Cambodian DHS is the latest available household survey data for Cambodia, findings should be interpreted with caution when referring to the current context as the data are 8 years old. Application of a causal hypothetical model to explain possible pathways to children's linear growth presents the first study of its kind using Cambodian DHS data, and lays the foundation for similar studies in other geographical contexts. However, the cross-sectional nature of DHS data and the use of structural path analysis, typically a form of causal modelling, require that the results be interpreted with some caution. Although structural path models may offer benefits over multivariable fixed effects models in allowing for complex interactions between variables, the theoretical model was built iteratively based on prior theory and thus is, by no means, an exhaustive representation of the pathways to children's HAZ.

## CONCLUSIONS

The theoretical structural path model presented in this study proposes a plausible illustration of the pathways associated with children's height for age in the Cambodian context. Overall, results confirmed that children's dietary diversity and continued breastfeeding status mediate the associations between socioeconomic status and linear growth, at least for pathways from maternal employment and household wealth. Analysis suggests

that participation in maternal employment presents a significant barrier to continued breast feeding beyond 6 months of age, emphasising a need for maternity legislation that supports optimal breastfeeding practices, in line with WHO recommendations that infants be exclusively breast fed for 6 months, followed by continued breast feeding in addition to complementary feeding up to 2 years of age.[25] Improvements to maternity legislation should also be inclusive of women working in the informal sector (of which 96% of employed women in Cambodia participate). Low and middle-income country contexts such as Cambodia, where there are distinct urban and rural differentials in the incidence of stunting and where rapid urbanisation has caused a shift in the geographical patterning of stunting, may warrant further investigation. Thus, future research may extend this analysis to further consider the moderating effect of urban/rural residence on the proposed structural path model using simultaneous path analysis.

**Contributors** CMH conceptualised the study, prepared the data set for research and conducted statistical analysis under the supervision of M-LN and SP. M-LN and SP contributed to the interpretation and revised paper for intellectual content. CMH is responsible for the overall content as the guarantor. CMH, M-LN and SP conducted the final review and revised the manuscript for submission.

**Funding** This research was carried out as a part of a doctoral research project, supported by the UK Economic and Social Research Council (grant number ES/J500161/1).

**Competing interests** None declared.

**Patient and public involvement** Patients and/or the public were not involved in the design, or conduct, or reporting, or dissemination plans of this research.

**Patient consent for publication** Not applicable.

**Provenance and peer review** Not commissioned; externally peer reviewed.

**Data availability statement** Data are available in a public, open access repository. Data are available in a public, open access repository. Available: https://dhsprogram.com/data/.

**ORCID iDs**
Chloe Mercedes Harvey http://orcid.org/0000-0002-3732-2973
Sabu Padmadas http://orcid.org/0000-0002-6538-9374

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
