## [Reviewer comments · BMJ Open]

ARTICLE DETAILS

TITLE (PROVISIONAL)	Maternal socioeconomic status and infant feeding practices underlying pathways to child stunting in Cambodia: structural path analysis using cross-sectional population data
AUTHORS	Harvey, Chloe; Newell, Marie-Louise; Padmadas, sabu

VERSION 1 – REVIEW

REVIEWER	Fernandes Nilson, Eduardo Augusto Ministry of Health of Brazil, Department of Health Promotion
REVIEW RETURNED	01-Dec-2021

GENERAL COMMENTS	The manuscript is based on the analysis of cross-sectional data (DHS) on maternal socioeconomic status and infant feeding practices underlying pathways to child stunting in Cambodia. The analysis of underlying factors that influence child growth is important to help understand contextual factors and their role, especially maternal factors such as education, employment, etc. Methods are clear and scientifically sound and results are well explored. In the discussion, it would be helpful to discuss the possible roles of food policy and food regulation on breastfeeding and child care. For example, the duration of maternal leave and the national incorporating of the International Code of Marketing of Breast-Milk Substitutes. Regarding the strengths and limitations, does the year of the DHS survey (2014) affect the timely analysis in terms of possible changes in the population after 6-7 years?
--

REVIEWER	Lydersen, Stian Norwegian University of Science and Technology, Regional Centre for Child and Youth Mental Health and Child Welfare
REVIEW RETURNED	20-Apr-2022

GENERAL COMMENTS	Figures 1 and 2 which are supposed to show the pathway models are missing in the manuscript. These figures are such crucial parts of the manuscript that I cannot carry out a review without access to these figures. On the contrary, two figure 1 on Page 26 and 28 seem to have nothing to do in this manuscript.
--

VERSION 1 – AUTHOR RESPONSE

Comments from Reviewer 1

C1: The manuscript is based on the analysis of cross-sectional data (DHS) on maternal socioeconomic status and infant feeding practices underlying pathways to child stunting in Cambodia. The analysis of underlying factors that influence child growth is important to help understand contextual factors and their role, especially maternal factors such as education, employment, etc. Methods are clear and scientifically sound and results are well explored. In the discussion, it would be helpful to discuss the possible roles of food policy and food regulation on breastfeeding and child care. For example, the duration of maternal leave and the national incorporating of the International Code of Marketing of Breast-Milk Substitutes. Regarding the strengths and limitations, does the year of the DHS survey (2014) affect the timely analysis in terms of possible changes in the population after 6-7 years?

R1: Thank you very much for your feedback and for your suggestion to include a discussion around the possible roles of maternity legislation and food policy and food regulation in the discussion. We have inserted some reflection on maternity legislation in Lines 396-401 and on food regulations in Lines 416-420.

Thank you also for noting the potential limitation due to the age of the data. The 2014 DHS represents the latest available data for Cambodia, however, we recognise the limitations that come with using older data and we have reflected on this in the strengths and limitations section (Lines 483-485).

Comments from Reviewer 2

C2: Figures 1 and 2 which are supposed to show the pathway models are missing in the manuscript. These figures are such crucial parts of the manuscript that I cannot carry out a review without access to these figures. On the contrary, two figure 1 on Page 26 and 28 seem to have nothing to do in this manuscript.

R2: Thank you for your comment and for indeed noting that the incorrect figures were included when the manuscript was uploaded. We apologise for this mistake and have ensured that the correct figures have been uploaded in this revision.

VERSION 2 – REVIEW

REVIEWER	Lydersen, Stian Norwegian University of Science and Technology, Regional Centre for Child and Youth Mental Health and Child Welfare
REVIEW RETURNED	20-Jun-2022

GENERAL COMMENTS	For the most part, you have used appropriate statistical methods with structural equation modelling and interpreted them the right way. But your study of possibly moderating effect of rural versus urban areas is partially erroneously reported. Please correct this, or delete this moderation analysis from the manuscript. My comment number 2 below is important. 1. But it is not always clear where you report constrained analysis (with respect to urban/rural areas) and where you report unconstrained analyses. Please state specifically above each table which is the case. I assume the unconstrained analyses are only in table 3.2. Page 15. Moderating effect of urban/rural residence.
---

	The conclusions here are not supported by the reported analyses. When you compare the effect of one association in rural areas compared to urban areas, it is erroneous to stat that there is a difference if $p < 0.05$ in one group and nonsignificant in the other. If you included the relevant interaction terms in the model, you could investigate if the effects differ. At least, you could provide confidence intervals for the coefficients in Table 3, and check to which degree they overlap. This would be better than what you have done, although that would not be a formal test either. 3. In general, I recommend reporting actual p-values, not just * or** or *** etc. Rather, give the actual p-value, for example $p = 0.12$ or $p = 0.035$ or $p = 0.006$. The exception is extremely small p-values, which ought to be reported as for example $p < 0.001$. I understand that this may be difficult in some of the tables and figures. But at least give the actual p-values in the text for example page 12 line 52 to 54 where you write “dietary diversity ($\beta = 0.07$; $p < 0.05$), ...” 4. Page 34 supplementary Figure S1 A bar chart is not suited for data which may have positive or negative values. I recommend another chart type, for example boxplot or a dot with two-sided error bar representing SD. 5. Page 10 line 17 You write “... whether there were statistical differences ... ” I suggest “... whether there were statistically significant differences ... ” 6. Table 1 and possibly elsewhere I agree that two decimals is a good choice of number of decimals for HAZ here. Pleas report 0 also when the last decimal(s) are 0, for example -1.60 not -1.6 in line 16. 7. Sex or gender You are not internally consistent. For example you use gender and sex in table 1. I recommend sex which is generally defined at the biological sex.
--	--

VERSION 2 – AUTHOR RESPONSE

BMJ Open: Maternal socioeconomic status and infant feeding practices underlying pathways to child stunting in Cambodia: structural path analysis using cross-sectional population data

MS reference no: bmjopen-2021-055853

REVISION NOTES

We are grateful to the Reviewers and the Editors for providing us with positive feedback and comments on our research paper. We have addressed each comment to the best of our abilities.

The revised text is amended in track changes. The following notes explain how we tackled each comment from the Reviewers. Please note C for comment and R for response.

C1: But it is not always clear where you report constrained analysis (with respect to urban/rural areas) and where you report unconstrained analyses. Please state specifically above each table which is the case. I assume the unconstrained analyses are only in table 3.

R1: Thank you for this comment. Indeed, table 3 presents the unconstrained analyses. However, as explained below, we have made the decision to exclude the additional analysis pertaining to hypothesis 3 which tests the moderating effect of urban/rural residence and we will re-do the analysis for a future extension of this paper.

C2: Page 15. Moderating effect of urban/rural residence.

The conclusions here are not supported by the reported analyses. When you compare the effect of one association in rural areas compared to urban areas, it is erroneous to stat that there is a difference if $p < 0.05$ in one group and nonsignificant in the other. If you included the relevant interaction terms in the model, you could investigate if the effects differ. At least, you could provide confidence intervals for the coefficients in Table 3, and check to which degree they overlap. This would be better than what you have done, although that would not be a formal test either.

R2: Thank you very much for bringing this to our attention, this is an important inaccuracy that you highlight. As you suggested we have decided to delete this moderation analysis from the current manuscript and develop an additional paper which will extend this analysis and present a more in-depth analysis of the moderating effect of urban/rural residence. As such, we have deleted this analysis where it appears in the manuscript and inserted two sentences in the conclusion to highlight our intention to conduct future research which considers this important aspect.

C3: In general, I recommend reporting actual p-values, not just * or** or *** etc. Rather, give the actual p-value, for example $p = 0.12$ or $p = 0.035$ or $p = 0.006$. The exception is extremely small p-values, which ought to be reported as for example $p < 0.001$. I understand that this may be difficult in some of the tables and figures. But at least give the actual p-values in the text for example page 12 line 52 to 54 where you write “dietary diversity ($\beta = 0.07$; $p < 0.05$), ...”

R3: Thank you for this point. Where it applies, I have changed the text to include the actual p-values, with the exception of p-values which are extremely small - in which case, they are reported as $p < 0.001$.

C4: Page 34 supplementary Figure S1

A bar chart is not suited for data which may have positive or negative values. I recommend another chart type, for example boxplot or a dot with two-sided error bar representing SD.

R4: Thanks for your recommendation – as per your earlier suggestion, we have removed this part of the analysis for this paper and will further develop the moderating analysis for a future paper which will build on the current paper.

C5: Page 10 line 17

You write

“... whether there were statistical differences ...”

I suggest

“... whether there were statistically significant differences ...”

R5: Thank you, this is an important clarification. However, since this part of the analysis has been removed from the current manuscript, it will be reflected in our future work.

C6: Table 1 and possibly elsewhere

I agree that two decimals is a good choice of number of decimals for HAZ here. Please report 0 also when the last decimal(s) are 0, for example -1.60 not -1.6 in line 16.

R6: Thank you for bringing this to our attention. I have amended the number of decimals in Table 1 and also checked that the same error does not appear elsewhere in the paper.

C7: Sex or gender

You are not internally consistent. For example you use gender and sex in table 1. I recommend sex which is generally defined at the biological sex.

R7: Thank you, this has now been amended in Table 1.

VERSION 3 – REVIEW

REVIEWER	Lydersen, Stian Norwegian University of Science and Technology, Regional Centre for Child and Youth Mental Health and Child Welfare
REVIEW RETURNED	06-Sep-2022
GENERAL COMMENTS	You have addressed my comments well.